# Promising Strategies to Reduce the SARS-CoV-2 Amyloid Deposition in the Brain and Prevent COVID-19-Exacerbated Dementia and Alzheimer’s Disease

**DOI:** 10.3390/ph17060788

**Published:** 2024-06-16

**Authors:** Nikita Navolokin, Viktoria Adushkina, Daria Zlatogorskaya, Valeria Telnova, Arina Evsiukova, Elena Vodovozova, Anna Eroshova, Elina Dosadina, Sergey Diduk, Oxana Semyachkina-Glushkovskaya

**Affiliations:** 1Department of Pathological Anatomy, Saratov Medical State University, Bolshaya Kazachaya Str. 112, 410012 Saratov, Russia; nik-navolokin@yandex.ru; 2Department of Biology, Saratov State University, Astrakhanskaya 82, 410012 Saratov, Russia; adushkina.info@mail.ru (V.A.); eloveda@mail.ru (D.Z.); ler.vinnick2012@yandex.ru (V.T.); arina-evsyukova@mail.ru (A.E.); 3Shemyakin-Ovchinnikov Institute of Bioorganic Chemistry, Russian Academy of Sciences, Miklukho-Maklaya 16/10, 117997 Moscow, Russia; elvod.ibch@yandex.ru; 4Department of Biotechnology, Leeners LLC, Nagornyi Proezd 3a, 117105 Moscow, Russia; eroshova.av@leeners.pro (A.E.); dosadina.ee@leeners.pro (E.D.); diduk.sv@leeners.pro (S.D.); 5Research Institute of Carcinogenesis of the N.N. Blokhin National Medical Research Center of Oncology, Ministry of Health of Russia, Kashirskoe Shosse 24, 115522 Moscow, Russia

**Keywords:** COVID-19, SARS-CoV-2 amyloid, Alzheimer’s disease, meningeal lymphatics, photobiomodulation, plasmalogens, natural compounds

## Abstract

The COVID-19 pandemic, caused by infection with the SARS-CoV-2 virus, is associated with cognitive impairment and Alzheimer’s disease (AD) progression. Once it enters the brain, the SARS-CoV-2 virus stimulates accumulation of amyloids in the brain that are highly toxic to neural cells. These amyloids may trigger neurological symptoms in COVID-19. The meningeal lymphatic vessels (MLVs) play an important role in removal of toxins and mediate viral drainage from the brain. MLVs are considered a promising target to prevent COVID-19-exacerbated dementia. However, there are limited methods for augmentation of MLV function. This review highlights new discoveries in the field of COVID-19-mediated amyloid accumulation in the brain associated with the neurological symptoms and the development of promising strategies to stimulate clearance of amyloids from the brain through lymphatic and other pathways. These strategies are based on innovative methods of treating brain dysfunction induced by COVID-19 infection, including the use of photobiomodulation, plasmalogens, and medicinal herbs, which offer hope for addressing the challenges posed by the SARS-CoV-2 virus.

## 1. Introduction

Alzheimer’s disease (AD) is a brain disorder that is accompanied by a gradual decline in memory, thinking and behavior skills. According to world statistics, the global number of people over 65 years old with AD doubles every 5 years [1]. As estimated, the incidence of AD will triple by 2060 unless the effective therapy is found to prevent or slow AD [1,2]. AD is the sixth leading cause of death in the United States [2]. During the period of 2020–2021, the number of deaths from stroke, cardiovascular disease, and human immunodeficiency virus decreased, whereas the number of people who died from AD increased more than 145% [2]. There is growing evidence that such a significant increase in mortality from AD may be associated with the COVID-19 pandemic in 2020 and 2021 [2,3,4,5,6,7,8,9,10,11,12,13,14,15,16,17]. COVID-19 is primarily known as a severe acute respiratory disease caused by the SARS-CoV-2 virus, but it also affects the central nervous system (CNS) and other organs [18,19,20,21,22,23]. Many neurological symptoms, including memory loss, cognitive impairment, fatigue, brain fog, and insomnia are reported in up to 30% of cases and may appear long after the infection (long COVID) [18]. However, despite considerable evidence that there is a relationship between COVID-19 and AD, it remains unclear if the SARS-CoV-2 virus itself causes AD and/or if the SARS-CoV-2 infection exacerbates existing AD. The virus hypothesis for AD has been proposed for decades, and despite supportive evidence, virus etiology of AD is still controversial [7,22,23,24,25,26,27,28]. On the one hand, many clinical observations indicate an increased risk for COVID-19 in people with AD [2,3,4,5,6,7,8,9,10,11,12,13,14,15,16,17]. On the other hand, SARS-CoV-2 infection causes long-lasting neurological sequelae [18,19,20,21,22,23]. The key question, whether COVID-19 can trigger new-onset AD or only accelerate AD progression, is unclear.

It was recently discovered that the SARS-CoV-2 virus proteome includes different sequences of amyloid-forming proteins [4,29,30]. It is assumed that amyloid-forming proteins from the SARS-CoV-2 virus may be involved in the development of neurological symptoms in COVID-19-infected patients [3,4]. The SARS-CoV-2 amyloid could have cytotoxic and neuroinflammatory effects similar to those of amyloids that are responsible for amyloid-related neurodegenerative diseases, including AD. Therefore, COVID-19-exacerbated AD could be a SARS-CoV-2 amyloid disorder. However, there is not enough evidence for this yet. Nevertheless, it has been noted that COVID-19 triggers systemic AA amyloidosis and upregulation of serum amyloid A protein [31]. On the other hand, amyloid beta (Aβ) has antiviral and antibacterial effects [32,33,34,35]. To this end, Aβ may be activated in the COVID-19-infected brain as a protective mechanism that can explain the proposed viral involvement in Aβ aggregation in the brain. However, despite the lack of a clear understanding of whether COVID-19 causes pathological amyloid accumulation in the brain or whether it aggravates already ongoing neurodegenerative processes, the indisputable fact is that the COVID-19 infection is accompanied by amyloid aggravation in the brain associated with the neurological disorders, including symptoms of AD and dementia.

Medicines may improve or slow the progression of AD [36,37,38]. However, there is no promising treatment that cures AD and reduces the accumulation of amyloids in the brain. Clinical investigations have failed to show any effective pharmacological therapy for AD [39,40,41,42]. The latest clinical trials of aducanumab on 3285 patients with AD revealed significant consequences associated with immunotherapy, such as edema, microhemorrhage, nausea dizziness, headache, and confusion [42]. However, Aβ immunotherapy can be improved by modulation of functions of the meningeal lymphatic vessels (MLVs) [43]. MLVs are now thought to be the major route for clearance of amyloids in AD [44,45,46,47,48,49]. Therefore, strategies for augmentation of lymphatic removal of amyloids from the brain can lay the foundation for the development of promising therapeutic approaches for AD [44]. MLV-based therapeutic strategies may be also useful for alleviating infection-induced neurological damage due to facilitation of virus clearance from the brain to the cervical lymph nodes [50].

However, despite the emerging evidence suggesting that MLVs may serve as an efflux route for Aβ from the human and animal brain, confocal analysis of sinus-associated MLVs in patients who died from AD has not found Aβ deposits directly inside of MLVs [51]. This suggests the need to develop alternative strategies beyond MLV modulation to stimulate amyloid clearance from brain tissue. Approaches using natural compounds as inhibitors of Aβ aggregation in brain seem to be most promising [52,53].

The focus of this review is to highlight new discoveries in the field of COVID-19-mediated amyloid accumulation in the brain associated with the neurological symptoms and the development of promising strategies to stimulate the clearance of amyloids from the brain through lymphatic and other pathways.

## 2. SARS-CoV-2 Amyloids and COVID-19-Mediated AD Dementia

Many proteins of viruses and bacteria have amyloid properties, which are thought to have an important role in the progression of infectious diseases [54,55,56,57]. The SARS-CoV-2 virus has four main structural proteins, namely the spike, envelope, membrane, and nucleocapsid proteins, as well as non-structural accessory proteins, including ORF6 and ORF10 [58,59] (Figure 1). The SARS-CoV-2 amyloids have been identified in the structural spike [60], the nucleocapsid [30], and the accessory ORF6/ORF10 proteins [4,61]. When the SARS-CoV-2 virus enters an infected cell, it creates many copies that are accompanied by an exponential increase in intracellular SARS-CoV-2 amyloids [62]. The spike proteins of the SARS-CoV-2 virus also bind to Aβ causing Aβ deposition in the brain [63,64,65] (Figure 1). The ORF10 protein causes mitochondrial dysfunction [66] mimicking AD-mediated metabolic disorders [4]. SARS-CoV-2 amyloids can be also released from infected cells into extracellular spaces, forming amyloid depositions leading to neurotoxicity and the pathophysiology of infection [4,30,60,62].

If the SARS-CoV-2 virus contains amyloid-forming sequences in its genome, the question arises, what is their role? It is known that every component of the virus helps it to impede the host immune system and replicate. Therefore, there are several possible roles of SARS-CoV-2 amyloids. The simplest explanation is SARS-CoV-2 amyloids can be an inflammatory stimulus [67] leading to the high expression of the angiotensin-converting enzyme 2 (ACE-2) receptors on the surface of microglia, through which the virus enters the cell, and that increases intracellular transmissibility of the SARS-CoV-2 virus. The nucleocapsid protein of the SARS-CoV-2 virus contains a lot of amyloidogenic sequences that may play an important role in RNA packing during virus replication [68,69,70]. It is also possible that SARS-CoV-2 amyloids might inhibit the host’s antiviral response, similar to other amyloids in other viruses [71].

Recent studies have reported that COVID-19 is associated with structural changes in the brain [72,73,74,75,76]. Therefore, severe COVID-19 patients demonstrate a decrease in grey matter volume of the frontal lobe [77,78,79]. The grey matter in the CNS allows for the control of movement, memory, and emotions [80,81]. Memory deficits have been found in 19.2% of post-COVID patients [78]. The SARS-CoV-2 virus can enter the hippocampus through invasion of the olfactory neurons [82,83]. Since the hippocampus plays a crucial role in regulation of episodic and spatial memory [84,85], hippocampal infection may lead to memory disorders in COVID-19 patients. Many forms of dementia are associated with deposition of different amyloids in the brain, raising the possibility that SARS-CoV-2 amyloids might be involved in the development of COVID-19-mediated dementia and AD or contribute to their progression through an increase of amyloid toxicity and aggregation [4,29,30,63,86,87,88,89,90,91,92].

There is also indirect mechanism of COVID-19-related AD dementia. COVID-19 as a respiratory diseases causes hypoxic changes to which the brain is highly sensitive [93,94,95,96]. It is noted that dementia and Aβ depositions are often accompanied by hypoxia of brain tissues [97,98,99,100,101,102,103]. This means that COVID-19-induced hypoxia might be an additional and independent factor promoting the development of dementia.

In sum, it has been discovered that SARS-CoV-2 amyloids play a potential role in COVID-19-mediated AD dementia [3,4,29,30,60,62,63,64,65,66] (Figure 1). By causing respiratory infection and systemic inflammation, the SARS-CoV-2 virus causes the development of a cytokine storm leading to blood–brain barrier (BBB) disruption [104,105,106,107,108,109]. We discussed the mechanism of COVID-19-related BBB damages in detail in our recent review [104]. The SARS-CoV-2 virus can enter the different regions of the brain through the damaged BBB as well as penetrating into the hippocampus through invasion of the olfactory neurons [82,83,84,85,110,111]. In the brain, the SARS-CoV-2 virus stimulates production of amyloid deposits in both extracellular and intracellular spaces [3,4,29,30,60,62,63,64,65,66]. The toxicity of SARS-CoV-2 amyloids can directly induce brain changes and lead to the development or the progression of AD dementia [4,72,73,74,75,76,78,82,83,84,85,112,113]. Thus, SARS-CoV-2 amyloids can play a causative role in COVID-19-exacerbated dementia and may be the targets for therapeutic approaches in prevention of COVID-19-mediated memory disorders.

## 3. Meningeal Lymphatics as a Target for Anti-Amyloid Therapy

Recent studies have demonstrated an important role of MLVs in the protection of the brain against pathogens and the development of neurodegenerative diseases [114,115,116,117]. Dysfunction of MLVs aggravates Aβ accumulation in the brain and significantly reduces the efficacy of immunotherapy against AD and brain tumors [43,44,118,119]. Several viruses, such as SARS-CoV-2, herpes simplex virus 1, rabies virus, Zika virus, vesicular stomatitis virus, and Japanese encephalitis virus, can invade the brain and induce inflammation with meningitis and neurologic symptoms [120,121,122,123,124,125]. It is known that the brain is protected against viruses, bacteria, and toxins by the BBB. However, viruses can enter the brain through the olfactory and trigeminal nerves pathways, bypassing the BBB [82,83,123,126,127]. It is important to note that MLVs can transport viruses from the brain to the peripheral lymphatics [114]. Li et al. revealed that that viral infection in mice reduces brain drainage and lymphatic removal of macromolecules from the brain. Photo or surgical damages to MLVs increase neurological symptoms and mortality in virus-infected mice. However, vascular endothelial growth factor C (VEGF-C) pretreatment promotes recovery of MLV functions against viral infection. These data indicate that MLVs facilitate lymphatic clearance of viruses, suggesting a promising MLV-based therapeutic approach against neurotropic viral infection [114]. Since MLVs play an important role in eliminating toxins and wastes from the brain, it is assumed that augmentation of MLV function could be a promising therapeutic target for preventing or delaying age-associated neurocognitive diseases, including COVID-19-mediated AD dementia [43,44,45,46,47,48,49,128,129].

Thus, the development of effective methods for stimulation of MLV functions is highly relevant for medicine. Recently, non-invasive transcranial photobiomodulation (PBM) has been proposed as a novel alternative method for therapy for dementia and AD as well as a promising approach in the improvement of cognitive functions in healthy subjects [128,129,130,131,132,133,134,135,136,137,138,139,140,141,142,143,144,145,146,147,148,149,150,151,152,153,154,155,156,157]. It is believed that the therapeutic effects of PBM are based on reducing neuroinflammation and activation of anti-oxidant systems through the improvement of metabolism and microcirculation of the brain, leading to neuroprotection [133,138,139,141]. However, emerging evidence suggests that PBM can also stimulate MLV functions, providing lymphatic removal of wastes and metabolites, including Aβ, from the brain, leading to reduced AD progression [151,152,153,154,155,156,157]. Notably, the U.S. Food and Drug Administration (FDA) recognized PBM as safe. The safe use of PBM is regulated by relevant standards, e.g., ANSI Z136 [158,159].

The mechanisms of PBM-mediated improvement of cognitive function in healthy subjects and with AD are not clear, but it is most likely that it may be due to PBM stimulation of MLV function. Indeed, disruption of MLVs aggravates AD, leading to cognitive deficit and behavioral alterations [44,47,48,49,50,160,161]. Thus, MLV dysfunction might cause cognitive decline and neurodegenerative disease due to aggregation of toxins in the brain, while PBM-stimulation of their lymphatic removal can improve the brain homeostasis, immunity and neuroprotection [43,44,46,47,48,49,128,129,130,131,132,133,134,135,136,137,138,139,140,141,142,143,144,145,146,147,148,149,150,151,152,153,154,155,156,157].

There is convincing evidence that PBM can modulate lymphatic contractility and pumping, improving drainage and removal of macromolecules from tissues (Figure 2a) [153,154,162]. The PBM-mediated increase in nitric oxide (NO) production could be one possible mechanism responsible for PBM-stimulation of the MLV functions. Indeed, PBM significantly increases NO generation in culture of isolated lymphatic endothelial cells [162]. The stimulating effects of PBM on MLVs are suppressed by the blockade of NO production [162]. How may NO promote contraction of lymphatic vessels? Traditionally, it is believed that NO suppresses lymphatic constriction. It is generally believed that NO causes relaxation of both blood and lymphatic vessels, suppressing their contractility. However, NO-related regulation of the lymphatic endothelium is much more complex than previously thought. It is known that the influx of fluid into the lymph vessel stimulates its contraction and NO production in the lymphatic endothelial cells increases in accordance with the intensity of vessel contractility [163]. Lymphatic constriction is accompanied by an elevated NO production in the lymphatic walls through an increased flow shear forces creating by coming fluids. In this case, NO causes dilation of the lymphatic endothelium after constriction, i.e., inhibits lymphatic pumping during high lymph flow. However, NO’s effects on different parts of the lymphatic vasculature are different. The lymphatic valves contain 50% more endothelial NO-synthase than tubular regions predominantly causing high NO production during the contraction of lymphatic vessels [163]. From a physiological perspective, during lymphatic constriction NO releases from the lymphatic valves into the lumen of the vessel and reaches tubular sites contributing their relaxation that is important for regulation of peristaltic pumping [163,164,165].

The contraction phase of the lymphatic vessels depends on the opening of the Ca^2+^-channels, while the relaxation cycle is a component of local production of NO in the lymphatic endothelial cells that triggers by transiently elevated shear forces (Figure 2b). Ca^2+^ and NO cooperate in their regulation of lymph flow via mechanobiological mechanisms. During the contraction cycle, shear stress is increased and triggers activation of the endothelial NO synthase and NO production in the lymphatic valves. In this case, the elevated production of NO reduces intracellular Ca^2+^ concentration, leading to preparation of the lymphatic endothelial cells for subsequent relaxation. During the relaxation phase, due to the reduced fluid velocity in the now dilated vessel, NO production drops, which is accompanied by its rapid degradation. At this time, intracellular Ca^2+^ levels and membrane potentials are restored in preparation for another contraction of the vessels, which can be triggered by any signal able to initiate the opening of Ca^2+^ channels or Ca^2+^ flux from calmodulin (calcium-binding messenger protein) or through the gap junctions of neighboring cells (Figure 2b).

We hypothesize that PBM can stimulate brain drainage and lymphatic clearance of macromolecules, including amyloids and viruses, from the brain due to PBM-mediated regulation of combination of mechanical and electrophysiological events associated with the pumping of the lymphatic vessels via PBM-mediated stimulation of NO production in the lymphatic endothelial cells (Figure 2c).

## 4. Plasmalogens as a Potential Therapy for AD

Lipids play a crucial role in the structure and functioning of the brain. The brain contains the second highest amount of lipids after adipose tissue. However, the type of lipids found in the brain is different from that in adipose tissue, with phospholipids being the main component in the brain. The brain contains a significant amount of mono- and polyunsaturated fatty acids such as oleic acid, arachidonic acid, and docosahexaenoic acid, which are mainly stored in phospholipids. These phospholipids are essential components of cell membranes and play a critical role in their proper functioning [166,167,168,169]. Cholesterol and phospholipids containing saturated fatty acids are concentrated in lipid rafts present in cell membranes. These rafts are involved in cell signaling. Membrane fluidity, supported by phospholipids consisting of unsaturated fatty acids, is crucial for membrane-associated functions. It includes the processing of amyloid precursor protein and plays a vital role in the pathogenesis of AD. The composition of phospholipids in cell membranes is involved in important neural functions, such as vesicular fusion and neurotransmitter release, which can be altered in AD [170,171,172,173].

Among phospholipids, plasmalogens play an enormous role in maintaining brain homeostasis. They appear to be the missing link between the biochemical and functional abnormalities observed in AD and its pathological signs, such as the accumulation of Aβ in the brain [174,175,176,177].

Plasmalogens are glycerophospholipids that are vital for brain function [178,179]. They are the primary structural components of lipoproteins, myelin, synaptic membranes, and cell membranes. Plasmalogens possess unique physicochemical characteristics that help regulate membrane fluidity, lipid packaging into lipoproteins, and interaction with nerve receptors and ion channels. These lipids are also necessary for synaptogenesis, myelination, and ion transport [180].

Plasmalogens are synthesized in the peroxisomes and endoplasmic reticulum. The initial reaction involves the acylation of dihydroxyacetone phosphate by acyltransferase. Cells tightly regulate plasmalogen levels. Excessive levels of plasmalogens act as feedback inhibitors, reducing the levels of fatty acyl-CoA reductase 1. This reduction occurs by promoting the degradation of protein, which provides fatty alcohols in the synthesis [179,181]. Peroxisome biogenesis disorders affect the metabolism of lipids and reactive oxygen species, leading to symptoms such as hypomyelination due to low plasmalogen levels and increased inflammation. Cells with a deficiency in plasmalogen synthesis have low oxidation resistance, but adding alkylglycerol as a plasmalogen precursor can increase plasmalogen levels and enhance oxidative resistance [176].

The emergence of age-related diseases such as metabolic syndrome, type 2 diabetes mellitus, Parkinson’s disease, and AD are associated with oxidative stress and chronic neuroinflammation. The disruption of redox homeostasis leads to peroxisome malfunction and impaired plasmalogen production, which could be a mechanism of the development of age-related diseases [174,182,183].

The level of plasmalogens in the brain increases until 30–40 years and then decreases sharply at about 70 years [184,185,186]. This age range coincides with the period of life when the incidence of AD increases exponentially [187,188,189]. Given the sharp decrease in brain plasmalogen levels with age and their critical role in maintaining brain homeostasis, it is not surprising that deficiency of plasmalogens in the brain is closely associated with the progression of aging-related neurodegenerative disorders such as AD and Parkinson’s disease [174,175,190].

Oxidative cleavage of the vinyl ether bond by cytochrome C in the presence of H_2_O_2_ [191,192] is a potential mechanism of degradation of plasmalogens. A decrease in peroxisomal function combined with higher H_2_O_2_ levels potentially causes a permanent plasmalogen deficiency, which leads to membrane changes, signaling abnormalities, reduced neurotransmission, and suppressed antioxidant protection [193]. Oxidative stress associated with inflammation potentially exacerbates the degradation of plasmalogens by attacking the vinyl–ether bond, further reducing anti-inflammatory and antioxidant capacity of tissues and initiating an irreversible vicious circle that progresses to pathological abnormalities [175].

During COVID-19 infection, the spike viral protein interacts with ACE2, which leads to the excessive production of angiotensin II and the activation of the nicotinamide adenine dinucleotide phosphate oxidase. This subsequently leads to an increase in oxidative stress and release of inflammatory molecules [194]. Unlike other viral infections, COVID-19 infection severity is usually associated with age and may be linked to an imbalance in the redox system. This can cause an accumulation of oxidative damage and a decline in the antioxidant defense system, resulting in an increase in reactive oxygen species [195].

Thus, plasmalogens, which reflect peroxisomes’ functional activity, can be used as biomarkers for diseases related to oxidative stress and aging, as well as a critical therapeutic target [182,196]. A study conducted by Yamashita et al. revealed that administration of Ascidian Viscera glycerophospholipid (EtnGpl), a plasmalogen-rich ethanolamine, to rats with an injected model of AD improves their learning ability and working memory through reducing oxidative stress in the brain [197]. Chronic injection of lipopolysaccharides in mice can lead to the activation of glial cells and accumulation of Aβ proteins in the brain that are associated with cognitive deficit. Hossain et al. clearly show that plasmalogens prevent neuronal cell death by activating protein kinase B and inhibiting caspase 3, improving neuronal survival [198,199]. Wood et al. found that the serum level of plasmalogens is significantly reduced in patients with AD and correlates with severity of dementia [200]. These data indicate that the circulating level of plasmalogens may be an indicator of the progression of AD.

There is growing evidence suggesting improvement of cognitive deficits in both animals and humans with neurodegenerative diseases. Thus, scallop- and ascidian-derived plasmalogens can reduce the incidence of cognitive impairment in subjects with mild forgetfulness, AD and Parkinson’s disease due to an increase in the plasma ethanolamine plasmalogen (PlsEtn) levels [201,202,203,204]. Chicken-derived plasmalogens have similar effects in healthy people with mild memory problems [205]. Phosphatidylserine plasmalogen species attenuate cognitive disorders after cerebrovascular injury [206].

The advantages of marine plasmalogens from ascidians, scallops, and sea cucumbers and the mechanisms underlying their beneficial effects on memory function are not well understood due to differences approaches in experiments, including dose, duration, and ethnicity of volunteers.

In aged mice, marine plasmalogens ameliorate memory deficit via activation of neurogenesis [53]. Amyloid-mediated neuroinflammation is associated with a decrease in the brain PlsEtn levels leading to suppression of the expression of the brain-derived neurotrophic factor (BDNF) and reduced neurogenesis [207,208]. These changes promote the high expression of p75 neurotrophin receptor (p75NTR) and protein kinase Cδ (PKCδ) leading to neural death and neurite degeneration [207,209]. However, the administration of PlsEtn reduces Aβ depositions in the brain and tau hyperphosphorylation by attenuation of the p75NTR and PKCδ expression, which leads to reducing neuroinflammation [207,209,210].

Chicken plasmalogens also enhance memory by attenuation of neuroinflammation [208,211]. However, marine PlsEtn enhances memory function better than chicken-derived plasmalogens, probably because marine PlsEtn is richer in DHA than the chicken derivative [208,210].

It is noted that brain levels of PlsEtn containing DHA in the cortex correlate with cognitive abilities of rats with AD [197,212,213]. In vitro experiments have shown that PlsEtn bearing DHA exhibits strong suppression of neuronal inflammation, apoptosis, γ-secretase activity, and Aβ aggregation [212,213,214]. Interestingly, intravenously injected small-sized liposomes can increase the bioavailability of PlsEtn with DHA, enhancing their beneficial effects on locomotor activity in normal rats [215,216].

Systemic administration of plasmalogens over 7 days reduces microglia inflammation induced by Aβ accumulation [217]. Longer treatment with scallop-derived plasmalogens over 15 months decreases both the expression of markers of inflammation and the protein kinase C-δ (PKCδ) involved in apoptosis [209]. Rothhaar et al. reported that murine plasmalogens reduce gamma-secretase activity in cell membranes isolated from post-mortem brain AD tissue [218].

However, murine plasmalogens can be beneficial for AD and improvement of cognitive function due to an increase in the brain PlsEtn level and neurogenesis, leading to reducing amyloid accumulation, tau hyperpolarization attenuating neuroinflammation and neuron death (Figure 3).

## 5. Natural Compounds as a Novel Treatment for AD

Current pharmacological therapy for AD includes use of cholinesterase inhibitors, anti-Aβ vaccine, and anti-neuroinflammation drugs [219,220,221]. Rivastigmine, galantamine, memantine, and donepezil are approved for treatment of AD [219,220,221]. However, these medications only partly improve quality of people with AD [222,223]. Monoclonal antibody anti-Aβ therapy, including aducanumab, lecanemab, and GV971, is a new strategy in therapy for AD in recent years. However, the safety and effectiveness of these drugs are still controversial, and these drugs are very expensive [224,225,226,227]. There are 43 new clinical drug trials registered in the National Library of Medicine database (ClinicalTrials.gov: accessed on 28 April 2024). However, the most common outcome was a lack of efficacy or only control of early symptoms of AD.

Thus, no effective cure for AD currently exists, and available pharmacological strategies have shown limited effectiveness. Therefore, the search for new approaches to AD therapy is crucial to address the growing burden of this pathology. Medicinal herbs are a promising avenue for the treatment of AD [228,229,230,231,232,233]. Natural compounds were the first therapy used for AD [233]. Although the mechanisms of therapeutic effects of ayurvedic medicinal herbs remain largely unknown, extensive experimental and clinical studies on phytochemical actions have found a wide range of beneficial effects for AD within these plants [228,229,230,231,232,233]. There are several natural components exhibiting therapeutic effectiveness for AD, including flavonoids, polyphenols, sterols, alkaloids, triterpenes, lignans, and tannins with effects against to oxidative stress, neuroinflammation, amyloid deposition, and activity of cholinesterase.

Here, we highlight the applications of various natural compounds as a novel treatment for AD (Figure 4).

### 5.1. Curcuma longa and Piper nigrum

Turmeric is the root of *Curcuma longa*, and its main active ingredients are curcuminoids. The most important and most abundant of these curcuminoids is curcumin [234,235]. Curcumin has been approved by the FDA as a natural dietary supplement due to its safety and non-toxicity. Scientists’ interest in curcumin increased due to its unique molecular structure, which has multiple effects, including anti-inflammatory, antioxidant, and neuroprotective actions [236]. It also has the ability to disaggregate tau proteins [237] and target amyloid plaques, which are one of the main causes of AD [238]. Studies have shown that among people aged 70–79, there is a 4.4-fold lower incidence of AD in India compared to the USA. This may be due to the fact that people in India consume more curry, which contains curcumin, and perform better on cognitive function tests [239].

Piperine is an alkaloid found in black pepper (*Piper nigrum*) and long pepper (Piper longum) [239]. It has been found that piperine can increase the absorption of various drugs [240], and it has also been observed that when curcumin is taken with piperine, the absorption of curcumin increases by 154% in rats and 2000% in humans [241,242]. It has been shown to improve the solubility, stability, and absorption of curcumin. However, curcumin has a poor oral bioavailability. To improve oral curcumin delivery to the brain, self-nanoemulsifying drug delivery systems (S-SNEDDS) have been developed using a nanotechnological approach [243]. In this study, the authors clearly demonstrate a significant dose-dependent therapeutic effect of combined curcumin with S-SNEDDS than drug alone in an AD model.

### 5.2. Ginkgo biloba

*Ginkgo biloba* is commonly used in the treatment of AD cognitive impairment due to its antioxidant and anti-apoptotic properties [244,245]. Its leaves contain various essential components, such as flavonoids, steroids, organic acids, ginkgolides, and terpenoids. These compounds, including bilobalide and ginkgolide, are crucial for its therapeutic effects [246,247]. Flavone glycosides make up approximately 22–27% of the extract, while terpene lactones comprise about 5–7% [246]. Specific terpene lactone compounds include A, B, and C ginkgolides and bilobalide.

*Ginkgo biloba* contains varies flavone glycosides, such as quercetin, isorhamnetin and kaempferol, that play a crucial role in preventing oxidative stress in AD [246]. The herb is thought to work through several mechanisms, including preventing apoptosis, reducing oxidative stress, inhibiting the formation of amyloid plaques, and scavenging free radicals. These actions help to prevent and treat AD. The plant extract also inhibits the neurotoxic effects of Aβ by regulating the activity of glutathione peroxidase and superoxide dismutase. This helps to limit neuronal apoptosis, reduce the build-up of reactive oxygen species, and improve glucose uptake and mitochondrial function. It also prevents the activation of the extracellular signal-regulated kinase and c-Jun N-terminal kinase signaling pathways. There is convincing evidence that *Ginkgo biloba* can enhance cognitive function in patients with AD by improving oxygen supply to brain tissues and reducing the brain level of free radicals [246,247].

### 5.3. Bacopa monnieri

*Bacopa monnieri* is a well-respected nootropic herb that has been used for centuries to treat neurological conditions. A chemical analysis of extract from *Bacopa monnieri* has revealed the presence of various bioactive compounds, including triterpenoids, alkaloids (such as nicotine, brahmine, and herpestine), saponins, glycosides, D-mannitol, herpaponin, monnierin, and alcohols. This plant also contains a variety of phytocompounds, such as bacosides A and B, bacosaponins A, B, C, bacosapeptides III to V, bacosapans D, E, F, jujubogenin, betulic acid, alkaloids, polyphenols, steroids, and sulfur compounds, all of which indicate its antioxidant properties. The antioxidant and neuroprotective effects of *Bacopa monnieri* make it a promising herb for AD. The main substances responsible for its neuroprotective effects are bacosides A and B due to their beneficial effects on the transmission of nerve signals and recovery of injured neurons [248]. The antioxidant activity of *Bacopa monnieri* also leads to an increase in various antioxidant molecules, including superoxide dismutase, thus reducing the harmful effects of H_2_O_2_-induced oxidative stress. In addition, this herb decreases lipoxygenase activity, assisting in the recovery from oxidative stress [249]. The diversity of bioactive compounds in *Bacopa monnieri* emphasizes its potential as a natural remedy for enhancing brain health and cognitive function.

### 5.4. Salvia officinalis

*Salvia officinalis* is a fragrant herb that is widely used in the treatment of AD [250,251]. Salvia contains more than 160 different polyphenolic compounds, including phenolic acids, and flavonoids. These include yunnaninic acid, lithospermum acids, salvianic acid, and others. It also contains flavonoids such as kaempferol, apigenin, and luteolin. In addition to these compounds, salvia is rich in terpenes such as alpha- and beta-thujones, 1,8-cineole, alpha-humulene, and camphor. These compounds can be found in the essential oils of the plant. Plants also contain significant quantities of diterpenes and triterpene acids, including tanshinones, carnosic acid, carnosol, and ursolic acid [249,250,251]. Salvia has great potential for promoting brain homeostasis and improving cognitive function. Indeed, *Salvia officinalis* has been shown to have anti-Aβ, anti-inflammatory and anxiolytic properties [249,250,251]. *Salvia officinalis* also demonstrates stimulating effects on the muscarinic and cholinergic pathways of the regulation of memory retention and formation, suggesting its supporting influences on cognitive properties.

### 5.5. Melissa officinalis

*Melissa officinalis*, also known as lemon balm, is a perennial herb with heart-shaped, bushy leaves that have a rough surface and grow upright [252,253]. The extracts from *Melissa officinalis* are rich in beneficial substances, including phenolic acids, flavonoids, and triterpenes. Thanks to these substances, the plant exhibits pronounced sedative, anti-depressant, and anti-inflammatory effects [254].

Studies have shown that lemon balm has the ability to inhibit acetylcholinesterase and exhibit antioxidant activity, making it valuable in preventing and treating AD [252,253,254,255,256,257]. In addition, lemon balm has anticholinesterase activity, binds to cholinergic receptors, and exhibits neuroprotective properties [256]. Some substances in lemon balm interact with muscarinic acetylcholine and nicotine receptors, leading to a decrease in acetylcholinesterase activity [256]. By modulating the cholinergic system, lemon balm is a promising treatment for AD, as it has been observed to reduce arousal and improve cognitive function in people with mild to moderate AD [252,253,254,255,256,257,258].

### 5.6. Huperzia serrata

*Huperzia A*, an essential alkaloid extracted from H. serrata, can effectively cross the BBB and acts as a reversible and selective acetylcholine esterase inhibitor demonstrating anti-AD effects [259,260].

### 5.7. Plants That Increase the Drainage Function of the Lymphatic System

MLVs play an important role in removal of wastes, metabolites, and toxins from the brain [44,45,46,47,48,49]. Therefore, MLVs are considered a promising target in therapy for brain diseases, including AD [43,44,45,46,47,48,49,128,129]. However, there are very limited pharmacological and non-pharmacological methods for augmentation of the MLV functions.

Recently, it has been proposed to introduce VEGF-C into the cisterna magna as a new method of pharmacological stimulation of MLV function [43,44,118,119]. However, this method has several limitations: (i) it is an invasive and traumatic method; (ii) VEFG-C has different side effects; and (iii) there are only few positive results in clinical studies in this area [261].

Herbal medicine has been proposed as alternative method in reducing lymphedema burden and drainage of different tissues [262,263]. Indeed, coumarin, juniper, black pepper, geranium, sage, and fennel are known to reduce swelling, and French oak wood extract containing robuvit leads to a sharp reduction in swelling [262]. Particular effectiveness of plant flavonoids has been shown in filarial lymphedema (edema of infectious origin) and in patients with breast cancer (edema of tumor origin) [264].

It is assumed that plant substances can enhance the drainage function of the lymphatic system via activation of the mechanisms lymphoneogenesis, including the high expression of VEFG proteins. Thus, Panax pseudoginseng increases mRNA levels of VEGF-C and vascular endothelial growth factor receptor-3 (VEGFR-3), leading to the stimulation and formation of new lymphatic vessels [265]. The stimulating effects on lymphangiogenesis have been found also for licoricidin, astragalus, and cinobufacini [263].

Plant substances can also be used to improve the drainage function of MLVs. Borneol (C10H18O, molecular weight 154.25) can quickly penetrate the BBB and enter the brain after oral administration within 5 min and reaches maximum concentration in the brain within 1 h [266,267], with the most efficient accumulation in the cerebral cortex and slightly less in the hippocampus and the hypothalamus [268,269]. Some studies demonstrate the effectiveness of borneol in therapy for AD, stroke, and epilepsy [270,271,272,273,274]. Indeed, Borneol improves removal of amyloids from the brain due to activation of lymphangiogenesis leading to improvement of cholinergic regulation of memory function.

## 6. Discussion

The recently discovered SARS-CoV-2 amyloids can play a crucial role in COVID-19-exacerbated dementia. Once the virus has penetrated the brain, it stimulates production of amyloid deposits at both extracellular and intracellular spaces. The toxicity of the SARS-CoV-2 amyloids can directly damage the brain, leading to the development or the progression of AD dementia. Therefore, SARS-CoV-2 amyloids may be targets for therapeutic approaches in prevention of COVID-19-mediated memory disorders and cognitive deficits. In this aspect, MLVs play a special role in the elimination of both amyloids and the SARS-CoV-2 virus from the brain [44,45,46,47,48,49,114]. It is actively being discussed whether increasing MLV functions will open a new page in the history of therapy for brain diseases associated with amyloidosis and lymphatic dysfunction, including AD, Parkinson’s disease, intracranial hemorrhages, brain trauma, and oncology [44,45,46,47,48,49,118,119,162,275,276]. However, this direction is in its infancy, which explains the limitation of technologies and pharmacological strategies for augmentation of MLVs to prevent or slow amyloidosis and MLV dysfunction. Among MLV stimulation technologies, the most promising is PBM because it is safe, without side effects, and approved by the FDA. There is emerging evidence suggesting that PBM effectively stimulates lymphatic removal of macromolecules, including Aβ and red blood cells from the brain as well as enhances brain’s drainage [153,154,155,156,157,162], leading to a significant improvement of memory function and cognitive properties in both healthy and AD subjects [128,157]. A pioneering direction is the development of portable technologies for PBM under the control of deep sleep. The animal and human data suggest that deep sleep is accompanied by strong activation of brain’s drainage and elimination of wastes and metabolites from the brain with the flow of brain fluids to the peripheral lymphatics [277,278]. It is obvious to assume that PBM of the MLV function and brain’s drainage will be most effective at the moment of their natural activation, i.e., during deep sleep [128,129]. Indeed, pilot results clearly show that PBM during deep sleep more effectively stimulates lymphatic removal of Aβ in an AD model and activates the brain’s drainage in both healthy mice and MLV-defective animals than PBM during wakefulness [45,128,129,151,157]. The use of PBM during sleep is also a promising direction in the treatment of brain tumors [279].

Among the pharmacological strategies for increasing the MLV functions, the direction of increasing lymphangiogenesis by introducing VEGF-C into the cisterna magna is widely discussed [43,118,119]. However, this method has limitations due to its invasiveness, side effects, and unproven clinical effectiveness. Therefore, the development of new alternative strategies to stimulate the MLV functions and remove macromolecules from the brain is a priority in medicine. Plasmalogens are a unique class of phospholipids playing an important role in membrane structure and in neural functions. Interestingly, the level of plasmalogens in the brain is dramatically reduced in AD subjects and is one of key mechanisms responsible for AD. The growing evidence demonstrates that plasmalogens can help to prevent some of AD progression through suppression of amyloid and tau accumulation, reducing neuroinflammation and activation of neurogenesis leading to significant improvement of cognitive and memory properties [53,197,198,199,200,201,202,203,204,205,206,207,208,209,210,211,212,213,214,215,216,217,218]. Plasmalogens also effectively increase resistance to chronic stress via activation of the immune system [280,281].

Large families of medical herbs have emerged as promising avenues in the treatment of AD [234,235,236,237,238,239,240,241,242,243,244,245,246,247,248,249,250,251,252,253,254,255,256,257,258,259,260,261,262,263,264,265,266,267,268,269,270,271,272,273,274]. Although the mechanisms of their therapeutic action remain largely unknown, extensive studies have identified various beneficial compounds in their extracts, including flavonoids, alkaloids, tannins, triterpenes with anti-inflammatory, anti-amyloidogenic, anticholinesterase, and antioxidant effects. These therapeutic effects suggest their potential as promising alternative treatments for AD and offer hope for addressing the difficulties posed by this incurable disease.

Given the new challenges associated with the rapidly growing problems of brain damage by the SARS-CoV-2 virus, immediate solutions aimed at increasing the resistance of the brain and its immune system to the COVID-19-mediated brain injuries are required. Since it takes 10 to 15 years for new pharmacological drugs to appear on the market, non-pharmacological methods of treating brain dysfunction induced by COVID-19 infection, including the use of PBM, plasmalogens, and medicinal herbs, should be more actively studied to develop effective alternative strategies to help COVID-19 patients.

## 7. Conclusions

Aβ is a normal product of neuronal metabolism eliminating from brain tissues through MLVs [44,45,128,129,151,157,277]. The conditions that can impair the function of lymphatic drainage processes, such as COVID-19 infection, or chronic sleep deprivation (less than 6 h), promote excessive accumulation of Aβ in brain tissues [3,4,29,30,128,129,282,283]. Recent findings have shown that SARS-CoV-2 virus-related accumulation of Aβ, which is identical to amyloids in AD, leads to the development of COVID-19-induced dementia [3,4,29,30]. Since virus-induced dementia develops quickly (in just a few weeks), it is logical to assume that this process may be reversible, as is the case with mild cognitive impairment [284,285,286,287,288,289]. Sleep deficiency over 20 years can lead to the development of dementia in 30% of cases, probably due to long-term Aβ deposition in brain tissues [283]. However, 70% of people experiencing chronic sleep deficiency do not develop dementia, indicating that they are either resistant to this process or it is reversible in them [283]. Thus, dementia and accumulation of Aβ in the brain associated with COVID-19 infection and sleep deficiency can be reversible and therefore treatable.

There is evidence that the dissolved form of Aβ, located in the perivascular spaces, is most toxic to neurons and synapses [290]. Soluble Aβ is rapidly cleared from the human brain within 1–2.5 h [291]. However, this clearance is impaired in AD and other forms of dementia, leading to Aβ aggregation in the brain. Because MLVs play an important role in draining brain tissues and removing toxins and metabolites from the brain, future treatments for reversible forms of dementia are expected to involve stimulation of lymphatic drainage processes. Given the growing evidence of the existence of lymphatic vessels directly in the human brain [292,293,294,295], and not only in its meninges [296,297], technologies for activating lymphatic drainage processes will contribute to progress in the treatment of a wide number of brain diseases associated with dysfunction of the cerebral lymphatic system, such as AD, Parkinson’s disease, brain trauma and oncology [44,45,46,47,48,49,118,119,162,275,276].

Recent animal studies have proposed PBM as an effective technology for stimulating the lymphatic clearance of Aβ and blood products [45,46,151,162,298,299]. However, PBM has limitations due to light scattering as it passes through the skull and the low doses used for photo-therapy [300]. Therefore, the use of PBM alone may not be sufficient to effective treat dementia, even if it is reversible. The alternative methods that could enhance the stimulatory effects of PBM may improve its therapeutic efficacy. In this review, we analyze the potential effectiveness of natural compounds, such as plasmalogens and medical herbs, to enhance lymphatic drainage processes in the brain. It is assumed that the combination of PBM and natural ingredients can become a reliable alternative in the treatment of various forms of reversible dementia.

Unlike moderate dementia, dementia associated with AD is irreversible, and there are no prospects for the emergence of new methods for completely curing this disease. However, PBM has been shown to be effective in pilot clinical trials in controlling progression of AD [128,134,135,136,137,140,141,146,147,148,149,150]. It is expected that while the search for a reliable and safe pharmacological treatment for AD is underway, the combination of PBM with natural products will help prevent the rapid progression of the disease, as well as increase the brain’s resistance to the toxic effects of Aβ.

## Figures and Tables

**Figure 1 pharmaceuticals-17-00788-f001:**
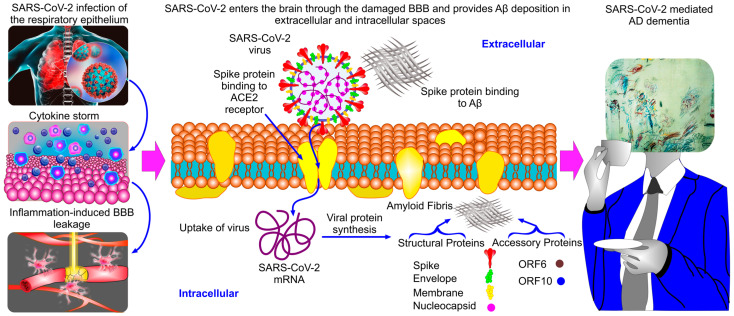
SARS-CoV-2 amyloid formation and the development of COVID-19-mediated AD dementia. SARS-CoV-2 causes respiratory infection associated with systemic inflammation and cytokine storms leading to BBB leakage. The SARS-CoV-2 virus can enter the brain through the damaged BBB and bind the ACE2 receptor. The SARS-CoV-2 virus can enter the hippocampus through invasion of the olfactory neurons. Once it enters the brain, the SARS-CoV-2 virus promotes the accumulation of amyloids both in the extracellular space due to the spike protein binding to Aβs, and intracellularly by forming the SARS-CoV-2 amyloids from the structural (the spikes, envelop, membrane and nucleocapsid) and the accessory (ORF6 and ORF10) proteins. These changes may lead to memory disorders in COVID-19 patients.

**Figure 2 pharmaceuticals-17-00788-f002:**
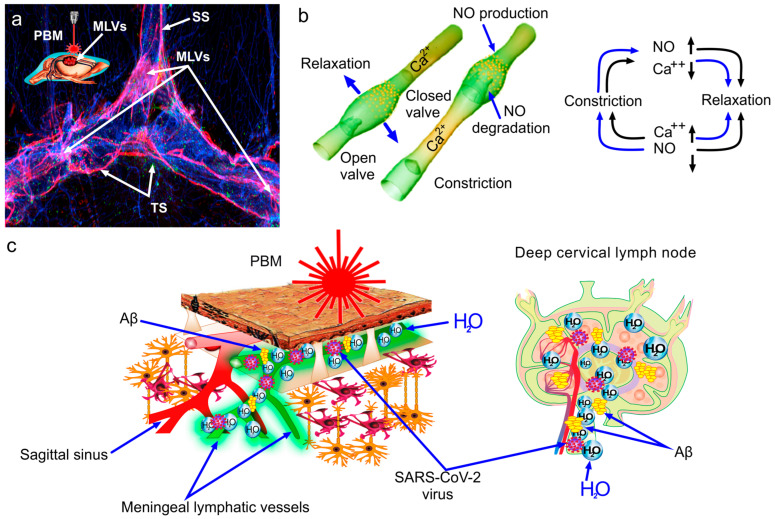
PBM-stimulation of lymphatic removal of the SARS-CoV-2 amyloids and virus from the brain: (**a**) Illustration of the effects of PBM on MLVs localized along the main venous sinuses, such as the Sagittal sinus (SS) and the transverse sinus (TS). (**b**,**c**) Schematic illustration of cooperation of NO and Ca^2+^ in regulation of lymphatic contraction and relaxation as well as the PBM-mediated stimulation of lymphatic removal of Aβ and the SARS-CoV-2 virus from the brain (detailed explanation in the text of the manuscript).

**Figure 3 pharmaceuticals-17-00788-f003:**
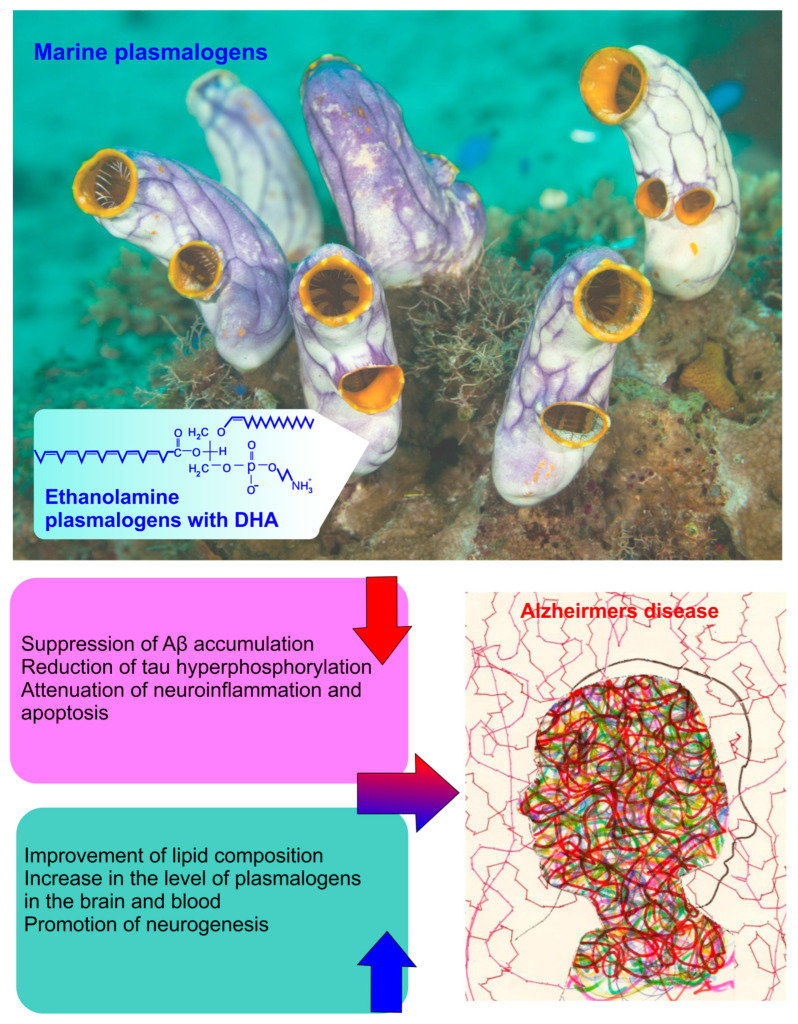
The beneficial effects of murine plasmalogens on cognitive function and AD.

**Figure 4 pharmaceuticals-17-00788-f004:**
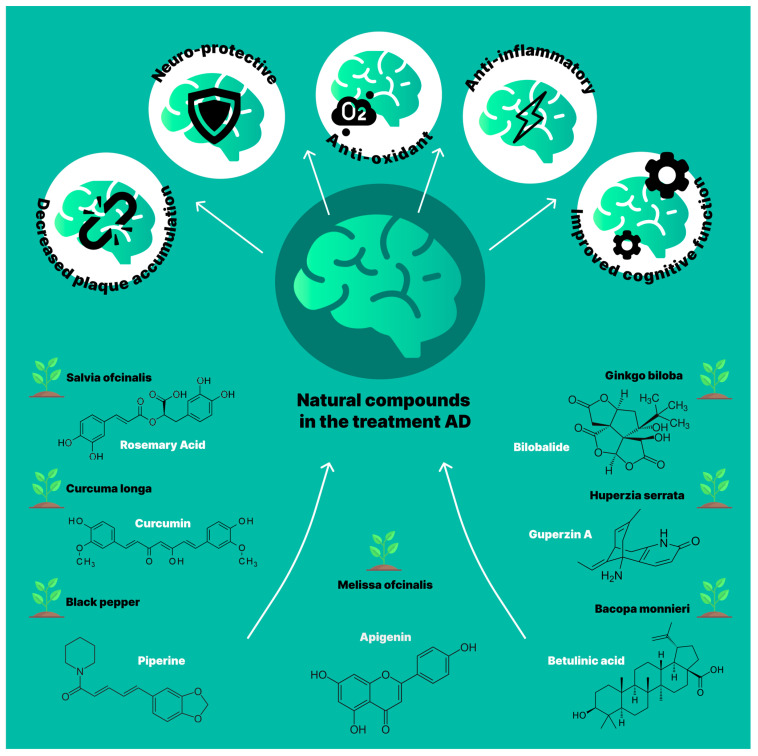
Schematic illustration of beneficial effects of promising plants for AD therapy.

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
