# Peer review of "Promising Strategies to Reduce the SARS-CoV-2 Amyloid Deposition in the Brain and Prevent COVID-19-Exacerbated Dementia and Alzheimer’s Disease"

_pharmaceuticals, 2024, doi:10.3390/ph17060788_

Round 1

Reviewer 1 Report

Comments and Suggestions for Authors

In this paper, the authors describe different therapeutic approaches to prevent the deposition of amyloid substances and suggest that these therapies could be used to combat brain damage associated with COVID-19. This approach is based on the fact that brain damage could be observed in COVID-19 with a deposit of Beta-amyloid substance. It is an unusual approach of COVID-19 disease, and the paper is illustrated by very beautiful figures.

However, the reviewer has enormous doubts about the effectiveness of the proposed therapies: in fact, in "classic" Alzheimer's dementias, the effectiveness of plants such as turmeric or ginkgo has not been proven. biloba to fight against disease. How can the authors imagine that substances without any real therapeutic effect could inhibit the very rapid deposition of amyloid substances induced by SARS-CoV-2 infection? this should be discussed. Overall, an explanation is really missing between the observation in some cases of amyloid deposition and SARS-CoV-2 infection.

Minor comment:

page 8, line 347: the term "ethnicity" would be preferable to "race"

Author Response

Comments: In this paper, the authors describe different therapeutic approaches to prevent the deposition of amyloid substances and suggest that these therapies could be used to combat brain damage associated with COVID-19. This approach is based on the fact that brain damage could be observed in COVID-19 with a deposit of Beta-amyloid substance. It is an unusual approach of COVID-19 disease, and the paper is illustrated by very beautiful figures.

However, the reviewer has enormous doubts about the effectiveness of the proposed therapies: in fact, in "classic" Alzheimer's dementias, the effectiveness of plants such as turmeric or ginkgo has not been proven. biloba to fight against disease. How can the authors imagine that substances without any real therapeutic effect could inhibit the very rapid deposition of amyloid substances induced by SARS-CoV-2 infection? This should be discussed. Overall, an explanation is really missing between the observation in some cases of amyloid deposition and SARS-CoV-2 infection.

Response: The authors express their sincere gratitude to the reviewer for the deep analysis of our review, constructive advices and recommendations. We send the revised article. All changes are highlighted in yellow. We added the discussion of how natural compounds may be effective in the treatment of AD in the form of combination therapy with PBM, as well as the relationship between COVID-19-induced dementia and the SARS-CoV-2 virus-related accumulation of Aβ in brain tissues (Lines 593-636; 1281-1326).

Comment: page 8, line 347: the term "ethnicity" would be preferable to "race".

Response: This is corrected (Line 247).

The authors once again thank the reviewer for the opportunity to improve the quality of our review and for its possible publication in the Pharmaceuticals.

Authors

Reviewer 2 Report

Comments and Suggestions for Authors

In my opinion, the manuscript is very interesting. It concerns the relationship between SARS-CoV-2 infection and neurodegeneration. The authors thoroughly analyze the immunological and molecular mechanisms that may underlie the increased risk of neurodegeneration associated with the content of sequences stimulating the formation of amyloid aggregates in the SARS-CoV-2 proteome. At the same time, they mention the antiviral and antibacterial properties of amyloid beta.

The authors' considerations are richly illustrated, making it easier to understand the issues discussed.

Additionally, in the second part of the work, the authors describe various compounds that may be useful to prevent neurodegeneration caused by SARS-CoV-2 infection. They describe in particular detail natural ingredients that can potentially be used in the adjuvant treatment of Alzheimer's disease. This part seems to be the least related to the main topic of the work and I ask the authors to consider abandoning this part or try to emphasize its connections with the main topic of the work.

Overall, the work is carefully written, well-organized and interesting. It makes a significant contribution to the development of knowledge regarding the links between COVID-19 infection and the risk of neurodegeneration. This is an important issue, which is why I believe the work is worth publishing.

Author Response

Comments: 

In my opinion, the manuscript is very interesting. It concerns the relationship between SARS-CoV-2 infection and neurodegeneration. The authors thoroughly analyze the immunological and molecular mechanisms that may underlie the increased risk of neurodegeneration associated with the content of sequences stimulating the formation of amyloid aggregates in the SARS-CoV-2 proteome. At the same time, they mention the antiviral and antibacterial properties of amyloid beta.

The authors' considerations are richly illustrated, making it easier to understand the issues discussed.

Additionally, in the second part of the work, the authors describe various compounds that may be useful to prevent neurodegeneration caused by SARS-CoV-2 infection. They describe in particular detail natural ingredients that can potentially be used in the adjuvant treatment of Alzheimer's disease. This part seems to be the least related to the main topic of the work and I ask the authors to consider abandoning this part or try to emphasize its connections with the main topic of the work.

Overall, the work is carefully written, well-organized and interesting. It makes a significant contribution to the development of knowledge regarding the links between COVID-19 infection and the risk of neurodegeneration. This is an important issue, which is why I believe the work is worth publishing.

Response: The authors thank so much the reviewer for taking the time to analyze our review and for its positive assessment. We send the revised article to which we have added, in accordance with the advices of the second reviewer, the discussion of how natural components may be useful for the treatment of AD as a combination therapy with PBM, as well as the relationship between COVID-19-induced dementia and the SARS-CoV-2 virus-related accumulation of Aβ in brain tissues. All changes to the text are highlighted in yellow.

Let us once again express our gratitude for a great help of the reviewer in the possible publication of our review in the Pharmaceuticals.

Authors

Round 2

Reviewer 1 Report

Comments and Suggestions for Authors

Thank you for your answers